# The Tolerance of *Eucalyptus globulus* to Soil Contamination with Arsenic

**DOI:** 10.3390/plants10040627

**Published:** 2021-03-25

**Authors:** Fernando Henrique Reboredo, João Pelica, Fernando C. Lidon, Maria F. Pessoa, Maria Manuela Silva, Mauro Guerra, Roberta Leitão, José C. Ramalho

**Affiliations:** 1Departamento Ciências da Terra, Faculdade de Ciências e Tecnologia, Universidade Nova de Lisboa, Campus da Caparica, 2829-516 Caparica, Portugal; fjl@fct.unl.pt (F.C.L.); mfgp@fct.unl.pt (M.F.P.); 2GeoBioTec, Departamento de Ciências da Terra, Faculdade de Ciências e Tecnologia, Universidade NOVA de Lisboa, Campus da Caparica, 2829-516 Caparica, Portugal; joaoferropelica@gmail.com (J.P.); abreusilva.manuela@gmail.com (M.M.S.); 3ESEAG-COFAC, Avenida do Campo Grande 376, 1749-024 Lisboa, Portugal; 4LIBPHYS, Departamento de Física, Faculdade de Ciências e Tecnologia, Universidade NOVA de Lisboa, Campus da Caparica, 2829-516 Caparica, Portugal; mguerra@fct.unl.pt (M.G.); rg.leitao@fct.unl.pt (R.L.); 5PlantStress & Biodiversity Lab, Centro de Estudos Florestais (CEF), Instituto Superior Agronomia (ISA), Universidade de Lisboa (ULisboa), Quinta do Marquês, Av. República, 2784-505 Oeiras e Tapada da Ajuda, 1349-017 Lisboa, Portugal

**Keywords:** arsenic toxicity, biomass production, *Eucalyptus globulus*, phytoremediation, photosynthesis tolerance

## Abstract

The contamination of abandoned mining areas is a problem worldwide that needs urgent attention. Phytoremediation emerges as a successful method to extract different contaminants from the soil. In this context, *Eucalyptus globulus* plants growing in soils artificial contaminated with arsenic (As) were used to access its phytoremediation capabilities. The effects of As on photosynthetic performance were monitored through different physiological parameters, whereas the uptake and translocation of As and the putative effects on calcium, iron, potassium, and zinc levels on plants were evaluated by X-ray fluorescence analysis. Root system is the major accumulator organ, while the translocation to the above-ground organs is poor. In the end of the experiment, the root biomass of plants treated with 200 μg As mL^−1^ is 27% and 49.7% lower than equivalent biomass from plants treated with 100 μg As mL^−1^ and control plants, respectively. Each plant can accumulate 8.19 and 8.91 mg As after a 6-month period, when submitted to 100 As and 200 As, respectively. It seems to exist an antagonistic effect of As on Zn root uptake by *E. globulus*. In general, the tested concentrations do not influence negatively plant metabolism, indicating that this species is suitable for plantation in contaminated areas.

## 1. Introduction

The contamination by metals and metalloids due to different anthropogenic activities is a reality worldwide [1,2] and in Portugal and a reason of concern of several scholars and public environmental agencies [3,4,5,6]. The mining activity was an important contribute to the Portuguese economy until the first half of the 20th century when the majority of the mines were closed [7]. In this context, huge tailings were formed with high concentrations of heavy metals and/or metalloids—arsenic (As), copper (Cu), lead (Pb), and zinc (Zn), with the consequent dispersion of contaminants mainly due to acid mine drainage [8,9].

Among the contaminants found in the biota, As is of particular interest especially in areas of the so-called Iberian Pyrite Belt (IPB), where the extraction of massive polymetallic sulfides occurred. Arsenic is present in more than 200 mineral species, the most common of which is arsenopyrite. In nature, As occurs primarily in its sulfide form in complex minerals containing silver (Ag), lead, (Pb) copper (Cu), nickel (Ni), antimony (Sb), cobalt (Co), and iron (Fe) [10]. Beyond the As origin from mining activities, the use of phosphate fertilizer during decades has been pointed as the main source of As contamination in Sri Lanka, with severe consequences to human health [11], although the use of commercial fertilizers derived from rock phosphates are also responsible for soil contamination with cadmium (Cd), leading the European Union to define three concentrations (60, 40, and 20 mg Cd kg P_2_O_5_) to limit further increasing Cd exposure on the long term in Europe [12].

Regardless the origin, As is a non-essential element and, generally, toxic to life. The contamination of ground water with As, used for irrigation and human consumption, or the consumption of rice or vegetables contaminated with As is a reason of concern worldwide [13,14,15]. The long-term exposure to inorganic As by humans can promote the triggering of bladder and lung cancer, as well as skin lesions [14], such as the so-called “*black-foot disease*” first reported in Taiwan in the 1960s due to the consumption of water from deep well with high As concentration [16].

The As toxic impact on plants is well documented ranging from the replacement of inorganic phosphate in biochemical reactions in the case of As(V), to the increment of reactive oxygen species, or the binding of As(III) to thiol reactive compounds [17]. The tolerance to very high concentrations of As is restricted to a few species (hyperaccumulators), although plants could be established in degraded mining soils with different degrees of bioaccumulation and tolerance [18,19].

In this context, some selected plants may have an important role in alleviating contaminant dispersion through an efficient root system uptake, as observed in some polluted salt marsh areas [20,21,22,23] or abandoned mining sites [24,25,26]. This mechanism, called phytoremediation, is a low-cost and efficient process that can contribute to a clean-up of contaminated soils. The success of phytoremediation depends on the growth rate of the plant and its biomass production, on plant uptake kinetics, beyond intrinsic soil characteristics [27,28], but without significantly compromising the plant metabolism. Tree species, due to longevity, large biomass accumulation and extensive rooting, might be more suitable for that purpose than herbaceous species, contributing for an efficient immobilization of the mineral contaminant [29].

The use of *Eucalyptus* sp. in phytoremediation purposes has been established with success in different world areas where the soils are contaminated with heavy metals [30,31]. Eucalypt plantations are widely spread in Portugal, managed in coppice systems (mainly 10–12 years rotations) for wood production used to feed the pulp and paper production. Currently, they occupy more than 800,000 ha in the mainland [32,33] with annual wood yields, ranging between 6 and 40 m^3^ ha^−1^, depending on the local conditions selected [34].

Plants living under natural conditions can be exposed to multiple stresses that interfere with photosynthesis, leading to growth decline and yield reduction. Thus, in vitro or in vivo conditions, the eco-physiological monitorization, is particularly recommended, using the photosynthetic pathway as probe of the global plant functioning, since it is, ultimately, responsible for plant biomass production and growth [35,36,37].

The aim of the current work was to assess the phytoremediation capability of *Eucalyptus globulus* Labill. (Tasmanian blue gum) grown in As contaminated soils during six months, in order to evaluate As uptake, accumulation and translocation efficiencies, and the effects on photosynthetic performance (as assessed through leaf gas exchanges, chlorophyll a fluorescence parameters, and biomass production), as well as the possible effects on the uptake of other important micro and macronutrients by the plant. This would allow to determine the potential for phytostabilization of soils from contaminated mining areas.

## 2. Results

### 2.1. Arsenic Accumulation

Using a portable X-ray Fluorescence apparatus (portable-XRF), As was only detected in the roots of *E. globulus* and the accumulation in this organ did not vary significantly in plants treated with 100 μg As mL^−1^ (100 As) throughout the experiment, i.e., from March till the end of July, with As concentrations ranging between ca. 49 and 54 μg g^−1^. Conversely, plants treated with 200 μg As mL^−1^ (200 As) exhibited greater values, with an increase in As concentration until May (102 μg g^−1^) decreasing afterwards. As expected, control plants are free of As (Table 1). When using a µ-Energy Dispersive X-Ray Fluorescence system (µ-EDXRF), it was possible to quantify As only in the leaves of As-treated plants, whose concentrations were not significantly different in March and July and were clearly lower than those found in roots (Table 1), the main accumulator organ.

Considering the whole As amount per plant, in March, 100 As and 200 As-treated plants showed similar values, despite the fact that 200 As plants showed a much lower biomass value. In May for close biomass values between As-treated plants, the 200 As-treated plants showed a 52% higher As concentration per plant than the 100 As plants, mainly due to the accumulation of As by the roots. However, in a longer term (July), the whole As accumulation within treated plants was similar, mostly due to an increase in root biomass of 100 As plants, whereas 200 As plants seemed to reached their maximal accumulation since no further accumulation occurred (Table 1).

Regarding the substrata, the accumulation of As in the soil reach a maximum of 36.4 μg g^−1^ in May with the contamination of 200 As, declining in July to 27.4 (in March a value of 32.7 μg g^−1^ was observed). Soils contaminated with 100 As exhibited lower levels, i.e., 15.8, 10.2, and 11.2 μg g^−1^, in March, May, and July, respectively.

### 2.2. Plant Growth and Foliar Traits

Specific leaf area (SLA) presented a reduction trend with time, which was somewhat more pronounced in As-treated plants. The leaf weight ratio (LWR) and leaf area ratio (LAR) also decreased along the experiment, with the lowest values observed always in July, regardless of treatments (Figure 1). By this time, 200 As plants presented significant lower SLA and greater LWR values as compared with untreated plants. Regarding plant height, similar values were found between treatments in all dates (Table 1). An increment in total plant biomass occurs throughout the experiment in both control and 100 As-treated plants, whereas biomass increment in 200 As plants remains stable from May to July.

### 2.3. Photosynthetic Related Parameters

Surprisingly, the net photosynthesis rate (P_n_) values were not significantly different regardless treatments and As exposition times, although 200 As plants consistently showed a tendency to lower values (Figure 2). As regards to stomatal conductance to water vapor (g_s_) and transpiration rate (Tr), a large increase in both parameters under summer conditions was observed in 200 As plants, whose values are significantly different (*p* ≤ 0.05) from the remainders (Figure 2).

In line with the absence of significant P_n_ changes in the As-treated plants, the chlorophyll (Chl) *a* fluorescence parameters showed only a few significant impacts from March until July (Table 2). In fact, 100 As plants did not significantly differ from control plants in all parameters and dates, with the single exception of a reduction in the regulated energy dissipation in PSII (Y_(NPQ)_) in May, associated with a tendency for a higher quantum yield of non-cyclic electron transport (Y_(II)_).

Notably, even under the greatest As exposure (200 As), plants showed a change in energy capture reflected in the reduced initial fluorescence (F_0_) in May and a reduction in Y_(II)_ (accompanied with a tendency to lower the photochemical quenching coefficient or q_L_) in July. Yet, the latter was fully compensated with a rise in Y_(NPQ)_, without a rise in the non-regulated energy dissipation in PSII (Y_(NO)_) processes. Except for these minor cases, all parameters reflecting the PSII photochemical performance, i.e., maximum PSII photochemical efficiency (F_v_/F_m_) and actual PSII photochemical efficiency of energy conversion (F_v_’/F_m_’), the estimates of photochemical energy use (Y_(II)_, q_L_), of photoprotective (Y_(NPQ)_), of non-photochemical quenching (q_N_), and of uncontrolled (Y_(NO)_) energy dissipation processes as well as the parameters reflecting the PSII photoinhibition status, i.e., the rate constant of PSII inactivation (F_s_/F_m_’) and the chronic photoinhibition index (PI_Chr_) were not modified under the maximal As treatment, always as compared with the control plants.

### 2.4. Minerals Accumulation

#### 2.4.1. Potassium

The two As treatments did not have an antagonistic effect on the absorption of K (with the exception in 200 As plants in roots, in March), as compared with the control plants. In addition, for all the treatments, K concentrations globally decreased from March to July in roots, exhibiting in the latter month 0.53%, 0.64%, and 0.73%, in control, 100 As, and 200 As plants, respectively (Figure 3).

The concentrations of K were higher in stems and leaves than in roots. The 200 As plants stood out for higher K values in stems and leaves (greater than 2%) by March, as well in July for leaves, always as compared with control plants. Potassium values were mostly maintained along the experiment in the stems, whereas in leaves, they tend to decline from March to May, being maintained afterwards (except in 200 As plants where an increase was observed).

#### 2.4.2. Calcium

In roots, Ca followed a somewhat opposite trend of K, i.e., decreasing, as As exposure increased, thus suggesting an antagonistic impact only in March (Figure 3). However, by May and July, no consistent differences were observed between treatments. In the stems and leaves, only 100 As plants showed a significantly increase in Ca concentration until May, but the 200 As plants did not differ from the control counterparts in any of the months. Altogether, minimum and maximum Ca concentrations were found in T3 in stems (*ca*. 2.6%) and in roots (*ca*. 4.3%) for 100 As plants, and in T1, in stems (*ca*. 2.1%) and T3, in leaves (*ca*. 4.4%) for 200 As plants. The minimum Ca content of the control plants was also observed in the stems (*ca*. 2.1% in T3), while the maximum value was observed in the leaves (*ca*. 4.5% in T3).

#### 2.4.3. Zinc

The highest levels of Zn were always observed in the roots and in control plants, what suggests an antagonistic effect of As on Zn, which was usually clearer in 200 As than in 100 As plants (Figure 4). Notably, maximal differences were observed in May, when the control plants showed 1.4 and 3.4 times more Zn than 100 As and 200 As plants, respectively. In July, such increases were only of 20% and 50%, in the same treatment order. Overall, stems and leaves showed as well significant Zn decline in As-treated plants. That trend was more pronounced in 200 As plants, whereas in 100 As plants, a much moderate (or absence of) impact was observed, compared with control plants.

#### 2.4.4. Iron

The accumulation of Fe in roots increased over the experimental period, with July values falling in the range between 1500 and 1600 μg g^−1^, similarly for all treatments. In fact, by March As-treated plants showed increases of 260% (100 As) and 200% (200 As) as compared to control. However, in May, only the 100 As plants still showed a significant greater value (70%) than control plants, and in July, no significant differences were found among treatments—the concentrations of Fe range between 1522 μg g^−1^ (200 As) and 1597 μg g^−1^ (100 As), while control plants exhibited an intermediate value of 1561 μg g^−1^ (Figure 4). In addition, an increase from May to July in all the treatments was observed in the leaves, although in July despite the differences in Fe concentrations among treatments, the mean values are not significantly different at the 0.05 significance level (Figure 4). Regarding stems, we cannot establish a trend of accumulation in this organ, although a peak was observed in May.

## 3. Discussion

### 3.1. As Accumulation, Photosynthesis, and Biomass Production

Similarly to other minerals and plant species, the uptake of As by *E. globulus* and the accumulation in plant tissues would depend on multiple interacting factors that include the plant/genotype itself, the local climate, and the soil characteristics [38,39].

When studying the accumulation of heavy metals by *Eucalyptus camaldulensis* living in nutrient-poor soils along the Guadiamar River valley (SW Spain) in seven different areas, strongly contaminated by a mine-spill in 1998, i.e., soils contained up to 1069 mg kg^−1^ of As and 4086 mg kg^−1^ of Pb, it was observed that the translocation of heavy metals to the leaves and flower buds is very small (the exception was Mn, which accumulated in *E*. *camaldulensis* leaves in two sampling points), thus indicating that this specie is suitable for phytostabilization of soils contaminated by heavy metals [40].

Other authors pointed out to higher accumulation by *Eucalyptus* species although it must be taken into account the methodology used. For example, in hydroponic culture, it was possible to reach a maximum of 315 μg As g^−1^ in the roots and 10 μg g^−1^ As in the leaves of *E. grandis x E. urophylla*, after 14 days exposure to 30 mg As L^−1^ in the form of Na_3_AsO_4_ [41]. In our case, using potted plants with solid substrate, As concentration in roots ranged from close to 50 μg As g^−1^, up to a maximum of 102 μg g^−1^, the latter in the 200 As plants (in May), when maximal As values were also observed in leaves.

A large variation in growth and contaminant uptake was assessed in 13 *Eucalyptus* clones, growing on agricultural field soils contaminated with As, Cd, Cr, Pb, Cu, and Zn [30]. The accumulation of As, Cu, Pb, and Zn was significantly higher in leaves than in stems and branches (in line with our finding), enhancing the potential for phytoremediation. In this trail, an average value of 2.9 μg As g^−1^ was observed in the leaves of 13 studied clones, (with a maximal value of 7.8 μg As g^−1^). Additionally, four clones revealed good phytoextraction performance [30].

The suitability of four *Eucalyptus* species (*E. cladocalyx, E. melliodora, E. polybractea, E. viridis*) for phytostabilization of As from gold mine tailings showed that all species accumulated, after 5 years growth, low As concentrations in the leaves, the highest levels being observed in mature leaves, ranging from 0.3 to 5.1 μg g^−1^ [42]. Although a great variation was observed within the species, *E. polybractea* had significantly higher foliar As values than the other species.

The *E. globulus* was able to triple the As concentration from March to May (10.8 μg g^−1^ in the 200 As plants by May)—Table 1. Notably, despite the high leaf accumulation (as compared with other above referred *Eucalyptus* species), *E. globulus* did not globally show significant changes in P_n_ (Figure 2), particularly in March and July. In fact, despite the tendency to a lower P_n_ when maximum As values were observed (200 As plants in May), associated with a decline in F_0_ (but not in F_v_/F_m_), no significant impact was found in the photosynthetic parameters related to PSII photochemical performance (F_v_’/F_m_’) and photochemical energy use (Y_(II)_, q_L_) (Table 2).

Furthermore, even when the unique impact on Y_(II)_ was observed (200 As plants in July) that was compensated by an increase in Y_(NPQ)_, reflecting the triggering of photoprotective dissipation mechanisms (which represented 71% of the total quantum yield estimates). Additionally, Y_(NO)_ remained quite stable (it even declined along the experiment in all treatments) similarly to what happens in other environmental stresses [43,44].

This estimate is associated with the non-photochemical quenching attributable to photoinactivation and non-regulated energy (heat and fluorescence) dissipation in PSII [43,45,46], and its low variability in As-treated plants points to the preservation of PSII functioning, as confirmed also by the absence of significant modifications in the parameters reflecting the PSII photoinhibition status (F_s_/F_m_’, PI_Chr_).

Our findings are in line with those for lettuce, where impacts on P_n_ (and g_s_) were observed for leaf As concentrations close or above 25 μg As g^−1^ (DW) [47] or with the finding regarding the hyperaccumulating populations of the brassica *Isatis cappadocica*, collected in As-contaminated mine spoils, which showed unaffected Fv/Fm and electron transfer rate values when exposed to As in the range of 5–800 μM, indicating normal photosynthetic efficiency and strength of plants in the presence of very high As levels [48].

Therefore, despite some minor impacts, including the tendency to a lower P_n_ and a large variability of values between leaves when maximum As values were observed, the photosynthetic machinery of *E. globulus* revealed tolerance at leaf level to the applied soil As levels. Such tolerance could be associated with greater antioxidant capability [49] and As detoxification mechanisms in both leaves and roots, as reported for *E. grandis* × *E. urophylla* [41]. These mechanisms might involve vacuolar compartmentalization, as well as the increase in sulfhydryl compounds (including non-protein thiols and glutathione, with antioxidant capabilities) and phytochelatins [41].

The halophytic shrub *Tamarix gallica* grew for 3 months with an irrigation solution supplemented with 0, 200, 500, and 800 μM As, under the form of sodium arsenate, and with and without 200 mM NaCl. The photosynthetic parameters Fo, Fm, and Fv fluorescence, as well as Fv/Fm ratio, were not affected by As nor by As combined with salt, occurring the same for pigment (chlorophyll a and b and carotenoids) and nutrient (K^+^, Ca^2+^, and Mg^2+^) contents. However, the highest As concentration, affected only the plant growth, but not the chlorophyll apparatus and the nutrient contents, yet [50].

In agreement with the low impact on photosynthetic performance, leaf biomass remained unaffected throughout the experiment. In fact, the biomass production was affected in the As-treated plants only in July, particularly in 200 As plants, and largely explained by the impact in root biomass (Table 1). As an initial exposed organ, it is usual that As inhibits root extension and proliferation, although most plants possess mechanisms to retain much of their As in the root system [35]. That seemed to be the case of *E globulus*, since root biomass was affected only after 7 months of harsh As exposure, clearly indicating that root development and consequent uptake efficiency can be affected by the highest As concentration used and exposure time.

### 3.2. Accumulation of Macro- (Ca and K) and Micronutrients (Fe and Zn) in As-Treated Plants

The four studied elements (Ca, K, Fe, and Zn) are essential to plant nutrition and the macronutrients Ca and K exhibited by far the highest concentrations in plants. The adequate concentrations in dry tissues are 0.5% and 1.0% for Ca and K, respectively, a value only surpassed by N with 1.5% [51]. Calcium is required for various structural roles in the cell wall and membranes, it is a counter-cation for inorganic and organic anions in the vacuole, and the cytosolic Ca^2+^ concentration is an important intracellular messenger [52]. Regarding K, it plays an essential role in enzyme activation, protein synthesis, photosynthesis, osmoregulation, opening and closing of stomata, energy transfer, phloem transport, ionic balance, and also stress resistance [53].

Iron and zinc are fundamental micronutrients in plants, the first one required for chlorophyll synthesis and component of cytochromes, while the second is involved as activator or component of several enzymes [51]. The adequate concentrations in dry tissues are 100 and 20 mg Kg^−1^ for Fe and Zn, respectively [51].

The recognition that As interferes with the balance of some nutrient in a species-dependent manner was pointed out by several authors for fava bean [54] and barley [55]. In *Vicia faba,* it was observed that all the concentrations of As (100, 200, and 400 µM) induced decrease on growth parameters, photosynthetic pigments, and mineral contents (N, P, K, Ca, and Mg) in the shoots, as compared to control plants, but increased lipid peroxidation, Na concentration, and total phenolic compounds [54]. Foliar application of 100 µM nitric oxide reversed the inhibition induced by As.

When examining the combined effect of soil-applied P (50, 100, 200, and 400 mg kg^−1^) and As (0, 30, and 60 mg kg^−1^) on elemental concentrations of sunflower plants under glasshouse conditions [56], it was observed that plant growth was significantly reduced with increased As supply regardless of applied P levels, and As toxicity caused significant decreases in the concentrations of K, Ca, Mg, Si, Fe, Zn, Cu, Rb, and Sr. On the other hand the uptake of Cu, Mn, Fe, and P by *Pistia stratiotes* increased until the concentration of 13 μM As, decreasing with higher concentrations (16 and 20 μM As), although no effects had been observed in the uptake of K, Ca, and Zn, regardless the concentrations of As, which lead the authors to consider the plant as an efficient As phytoremediator [49].

For the *Eucalyptus* genus Foelkel [57] reported in the leaves, K levels between 0.5% and 1.2% and Ca levels ranging between 0.4% and 1.0%. Furthermore, Merino et al. [58] report 0.6% of K and 0.32% of Ca in *E. globulus* leaves from field-grown experiments, pointing as well that great differences can be observed whether we monitored nutrients in plantations, pot cultures, or hydroponic ones. When evaluating the effects of different As levels (2 and 5 mg As L^−1^ added as sodium arsenite—NaAsO_2_) on *Phaseolus vulgaris* L. cv. Buenos Aires, a decline in K, Na, and Mg concentrations and increase in N and Ca levels was observed in roots [59].

Also, the effects of As (in the form of sodium arsenate) on the distribution of macronutrients in the roots and shoots of *Wrightia arborea* revealed that the accumulation of macronutrients (Ca, K, Mg, P, and S) varied differently under As exposure. In the case of K, As concentrations between 0.2 and 2.0 mM significantly increased K content in the roots, but did not affect shoot concentration [60].

Our findings showed that As did not reduce the K uptake by the roots nor the translocation to the above-ground organs in *E. globulus*—in the end of the experiment, both roots and leaves of 200 As plants showed a K content higher than the controls (Figure 3). Given the role of K in the ion balance, it is likely that such increased values in root and leaves will serve to balance the entry of anions resulting from excessive As uptake [61].

In the case of Ca, no significant changes were found in the end of the experiment, between control and 200 As plants—except in roots by March (Figure 3). The absence of a negative impact on Ca was in line with the overall maintenance of the parameters associated with PSII efficiency and photochemical use of energy (Table 2) and, globally, with the absence of negative impacts on P_n_, clearly in March and July (Figure 2). That is related with the well-known crucial role of Ca in the regulation of the photosynthetic pathway, namely, in processes such as, stomatal closure, photosystems performance, non-photochemical quenching, and xanthophyll cycle [62,63]. Furthermore, this absence of clear impact on Ca uptake (as well as on K) is in line with other plants considered with high phytoremediation potential [49].

As regards to the impact of As on Zn contents, it was reported that leaves of *E. camaldulensis* and *E. globulus* plants growing in heavily As contaminated soils, near deactivated mines, accumulated ca. 80 and 25.5 μg Zn g^−1^, respectively, with As values of 0.6 and 0.3 μg g^−1^, in the same species order, suggesting differences between species and mechanisms of tolerance [18]. Such value obtained for *E. globulus* was somewhat lower than the values obtained by the current study in As-treated plants.

Furthermore, it is noteworthy that in the end of the experiment, the leaf Zn content is similar in both 100 and 200 As plants, even when leaf tissues contained 6.7 and 4.4 μg As g^−1^, respectively, a much greater As concentration range than those reported by [18]. Such harsh As environment would justify the clear antagonism with Zn, particularly when comparing the 200 As plants with their controls as regards to all plant tissues (although not significantly in leaves)—Figure 4. A similar finding was noted in *Spartina alterniflora* when submitted with different concentrations of As^3+^ and As^5+^ [64]. In general, it was observed a marked decrease in Zn content in the roots of the plant, regardless the concentrations applied, while the concentration in the shoots was always higher than those found in control plants [64].

The Fe concentrations observed in *E. globulus* organs are in any case much higher than the concentration considered adequate for plant tissues (on a dry weight basis), which is 100 μg g^−1^ [51]. The accumulation of Fe in roots as response to As toxicity was evidenced by Shaibur et. al. [55] who showed that the roots of barley shoots (*Hordeum vulgare*) accumulated 262.6 μg Fe g^−1^ after treatment with 10 μM, while As-free plants contained only 24.3 μg g^−1^. Conversely, aerial concentrations were higher in control plants compared to treated plants (63.8 vs. 26.3 μg g^−1^) [55], with a similar pattern of decline observed in the leaves of *E. globulus* by July. Globally, it seems that As did not significantly changed the Fe content in the roots by the end of the experiment, although the effect on the leaves was more pronounced on 200 As plants—207 vs. 329 μg Fe g^−1^ in control plants.

Regarding the efficiency in the uptake of As from contaminated soils and translocation to the above-ground organs, we must emphasize that the whole biomass in March (roots and leaves) accumulated a total amount of 2.5 and 2.4 mg As, in 100 As and 200 As-treated plants, respectively, whereas in May and July, plants treated with the highest concentration accumulated close to 10 and 9 mg As, respectively. With the lowest concentration, it was possible to detect a total As load of 8.2 mg in July.

In this context, the As load removed within a short period may be significant especially in adjacent areas of mining activities, although as previously referred [18], the leaves of *Eucalyptus camaldulensis* and *Eucalyptus globulus* growing in heavily contaminated soils around a deactivated mine exhibited foliar As concentrations up to 0.6 μg g^−1^, a level sixfold lower our minimum value, thus indicating that “*all the extrapolations from plant behavior* in vitro *must be avoided or done with great care*” [65]. Nevertheless, the use of *E. globulus* in abandoned mining areas seems to be adequate for phytoextraction, although the recovery from the soil would be expected to be slower than assays in both soil and hydroponic cultures.

## 4. Materials and Methods

### 4.1. Plant Material and Experimental Design

Five months old *Eucalyptus globulus* Labill. plants were collected from Altri Florestal S.A. and transplanted to 5 L pots, with 3 L of subtract SIRO Universal/Portugal (pH: 5.5–6.5; humidity: 50–60%, electrical conductivity: 0.6–1.2 (mS m^−1^), N: 80–150 mg L^−1^; P_2_O_5_: 80–150 mg L^−1^—corresponds to 35 and 65 mg P; K_2_O: 300–500 mg L^−1^—corresponds to 249 and 415 mg K; organic matter: >70%). Plants were then acclimated during 3 months (from September to December) to natural conditions in FCT/UNL, Campus da Caparica, Portugal (GPS—38°39′41,5″ N, 9°12′24,0″ W). The mean air temperature was 16.0 °C, and the average of maximum and minimum temperature were 21.9 and 10.1 °C, respectively [66]. The mean values of total annual rainfall reached 600 mm.

Plant distribution was randomized to obtain three groups of 24 plants each, constituting the control and two As treatments. Arsenic was added to the soil in January as NaAsO_2,_ soluble in 100 mL distilled water in two distinct concentrations, 100 (100 As) or 200 (200 As) µg As mL^−1^, while control plants received the same volume of distilled water only. After soil contamination in January (T0 moment), plant and soil analysis were carried every 2 months, i.e., March (T1 moment), May (T2 moment), and July (T3 moment).

### 4.2. Growth Analysis

Along the experimental period, several growth parameters were evaluated: the specific leaf area ratio (SLA), the leaf weight ratio (LWR), and the leaf area ratio (LAR) [67].

SLA = A/W_L_ (A = leaf area; W_L_ = leaf biomass)

LWR = W_L_/W (W_L_ = leaf biomass; W = total plant biomass)

LAR = A/W ((A = leaf area; W = total plant biomass).

### 4.3. Leaf Gas Exchanges Analysis

Net photosynthesis (P_n_), stomatal conductance to water vapor (g_s_), and transpiration rate (Tr) were obtained under photosynthetic steady-state conditions after, at least, 2 h of illumination (performed between 10:00–11:00 a.m., where preliminary evaluations showed to be the daytime with higher P_n_ and g_s_ values), using a portable open-system infrared gas analyzer (CIRAS 3, PP Systems, USA). Measurements were done by 1st April, 2nd June, and 30th July, i.e., after *ca*. 2, 4, and 6 months of soil contamination, under natural conditions, with an irradiance of *ca*. 1500 μmol m^−2^ s^−1^ and providing an external CO_2_ of *ca*. 390–400 μL L^−1^.

### 4.4. Chlorophyll a Fluorescence Parameters

Chlorophyll (Chl) *a* fluorescence parameters were evaluated on the same dates (environmental conditions) and leaves used for gas exchange measurements, using a PAM-2000 system (H. Walz, Effeltrich, Germany), as previously described [68], and following the formulae and meanings discussed elsewhere [44,45,69]. Measurements in dark-adapted leaves included the F_0_ (minimum fluorescence from excited Chl *a* molecules from the antennae) and F_v_/F_m_ (maximal PSII photochemical efficiency). A second set of parameters, evaluated under photosynthetic steady-state conditions (*ca*. 1500 μmol m^−2^ s^−1^ of natural irradiance) and superimposed saturating flashes (*ca*. 8000 μmol m^−2^ s^−1^), included the F_v_’/F_m_’ (PSII photochemical efficiency of energy conversion under light exposure), q_L_ (photochemical quenching based on the concept of interconnected PSII antennae, representing the proportion of energy captured by open PSII centers and driven to photochemical events) [44,69], and F_s_/F_m_’—predictor of the rate constant of PSII inactivation [70].

Additionally, estimates of photosynthetic quantum yields of non-cyclic electron transfer (Y_(II)_), photoprotective regulated energy dissipation of PSII (Y_(NPQ)_), and non-regulated energy dissipation of PSII as heat and fluorescence (Y_(NO)_), where [Y_(II)_ + Y_(NPQ)_ + Y_(NO)_ = 1], were also calculated [44,71]. Finally, it were evaluated the PSII photoinhibition indexes [46,72] that included: (A) chronic photoinhibition (PI_Chr_), representing the percent reduction in F_v_/F_m_ at each temperature relative to the maximal F_v_/F_m_ obtained during the entire experiment; (B) dynamic photoinhibition (PI_Dyn_), representing the decline in F_v_/F_m_ that is fully reversible overnight, being measured as the percent reduction in midday F_v_’/F_m_’ relative to F_v_/F_m_ at each temperature, relative to the maximal F_v_/F_m_ from the entire experiment; and (C) total photoinhibition (PI_Total_).

PIchr = [(*F*_v_/*F*_m_) max—(*F*_v_/*F*_m_) pd]/(*F*_v_/*F*_m_)max × 100%

PIdyn = [(*F*_v_/*F*_m_) pd—(*F*_v_/*F*_m_) mid]/(*F*_v_/*F*_m_)max × 100%

PI_Total_ = PI_Chr_+PI_Dyn_.

### 4.5. Metal Determination in Soil and Plant Samples

For determination of the As content, plants were removed from the soil, washed with tap water, and separated in roots, stems and leaves, which were then dried at 60 °C until a constant weight. Soils were also dried in a process similar to that previously described, before being analyzed. Thereafter, plant samples were powdered and stored.

Arsenic analyses in soil, roots, stems, and leaves were performed in triplicate using an X-ray Analyzer (Thermo Scientific, Niton model XL3t 950 He GOLDD+, USA). This model was also used to analyze some macronutrient (Ca and K) and micronutrients (Fe and Zn) in plant samples, in accordance with Environmental Protection Agency (EPA) method 6200 [73]. Detection limits using the optimum “mining” mode for a period of 120 s under high-purity helium (He) were As = 5 mg kg^−1^, Fe = 25 mg kg^−1^, Zn = 6 mg kg^−1^, Ca = 350 mg kg^−1^, and K = 500 mg kg^−1^. Soil reference materials (NRCan Till-1) were run before the beginning of analyses and after every five samples; the recovery values ranged between 93% and 98%.

### 4.6. Micro-Energy Dispersive X-ray Fluorescence (µ-EDXRF)

In order to overcome the limit detection of the portable XRF, plant organs were also monitored for As determination using a Micro-Energy Dispersive X-Ray Fluorescence (µ-EDXRF) Bruker M4 Tornado^TM^ system.

For analysis of the As content of *E. globulus* stems and leaves, dehydrated samples were previously reduced to powder in a mortar and transformed into pellets of 2 cm in diameter and 1 mm thick. This pellet was then glued onto a mylar sheet in a plastic frame and placed directly onto the X-ray beam for analysis. From each sample, three pellets were made and submitted to analysis. The As evaluation was determined with the µ-EDXRF equipped with a Rh-anode X-ray tube powered by a low-power HV generator. This equipment features a policapillary optics that allow a 25 μm in diameter that can be positioned anywhere on the sample. Detection was performed with a XFlash^®^ SDD detector with an energy resolution of 145 eV at an X-ray energy of 5.9 keV.

Due to the possible heterogeneity of the samples at the featured spatial resolution, an area of around 50 mm^2^ was scanned and the resulting total spectrum was analyzed, in order to get an average value of the concentration in that region. The mapping parameters featured a 15 µm pixel spacing and an acquisition live time of 6 ms per pixel.

Quantification of the total spectra of the obtained maps was performed with the fundamental parameters method of the built-in ESPRIT software and the recovery rate was checked against a set of standard reference materials—Orchard Leaves (NBS 1571) and Sea Lettuce (BCR-279), which were chosen given their matrix similarity with the samples and due to the As certified levels being both in the expected high concentration region of the samples (NBS 1571) and close to the instrument lower detection limit (BCR-279). The achieved detection limit with this setup can be seen in reference [74,75], is around 3 µg/g. The recovery values for As were 115% for the Orchard Leaves (NBS 1571) and 99% for the Sea Lettuce (BCR-279). In both SRM, the certified value was within the measured average ± standard deviation concentration interval.

### 4.7. Statistical Analysis

Data were statistical analyzed with comparison of means with one-way ANOVA, followed by a Tukey’s test for multiple mean comparisons and Pearson correlations using the SPSS statistical package (Version 14.0). A 95% confidence level was adopted for all tests. Elemental analysis was performed in triplicate (n = 3), whereas SLA, LWR, and LAR (n = 4); P_n_, g_s_, and Tr (n = 5); and finally, chlorophyll a fluorescence (n = 8).

## 5. Conclusions

The *E. globulus* can accumulate 8.19 and 8.91 mg As within the period tested, when submitted to 100 and 200 μg As mL^−1^, respectively, and taking into account the biomass produced and the measured As concentrations in the roots and leaves. Roots are by far the major accumulator organs, while the translocation to the above-ground organs is poor, in the case of the *E. globulus* stems, As if present, is below the detection limits of the µ-EDXRF system. In the end of the experiment, root biomass of 200 As plants greatly declined, but leaf biomass remained unaltered throughout the experiment. This indicates that root development and consequent uptake efficiency can be affected by the highest As concentration used and time of exposure, despite the maintenance of leaf biomass and photosynthetic performance.

Specific leaf area (SLA) exhibited a reduction trend with time, which was somewhat more pronounced in 200 As-treated plants. A similar pattern was noted for the leaf weight ratio (LWR) and leaf area ratio (*LAR*), with the lowest values observed always in the end of the experiment, regardless of treatments, when temperature rise in July. Despite this remark, 200 As plants presented greater LWR values as compared with control plants thus indicating that above-ground biomass production is not affected, whereas root biomass is strongly affected by As, compared with control plants.

Regarding possible interactions between As and micro- and macronutrients, it seems to exist an antagonistic effect of As on Zn root absorption by *E. globulus*, whereas the translocation of Fe to leaves is also affected by increasing concentrations of As. Altogether, our findings indicated that *E. globulus* can be suitable for plantation in contaminated areas, although the recovery from the soil would be expected to be slower than assays in both soil and hydroponic cultures.

## Figures and Tables

**Figure 1 plants-10-00627-f001:**
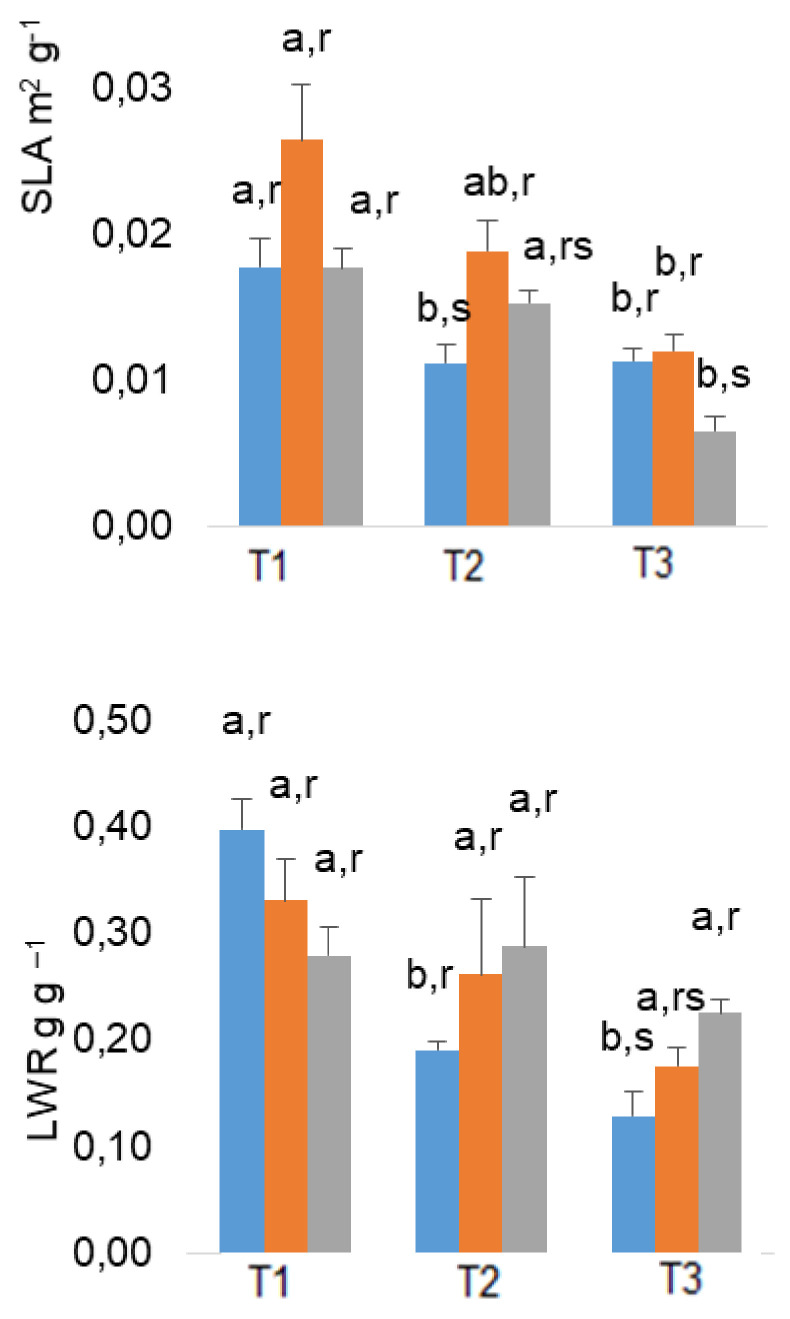
Variation in *specific leaf area (SLA)* leaf weight ratio (LWR) *and leaf area ratio (LAR),* in *E. globulus* plants, throughout the experimental (March; T2—May; T3—July). The mean values ± standard error (n = 4) followed by different letters express significant differences over time (a, b) or between As treatments (r, s), for the control (**blue**), 100 As (**orange**), and 200 As (**grey**) treatments.

**Figure 2 plants-10-00627-f002:**
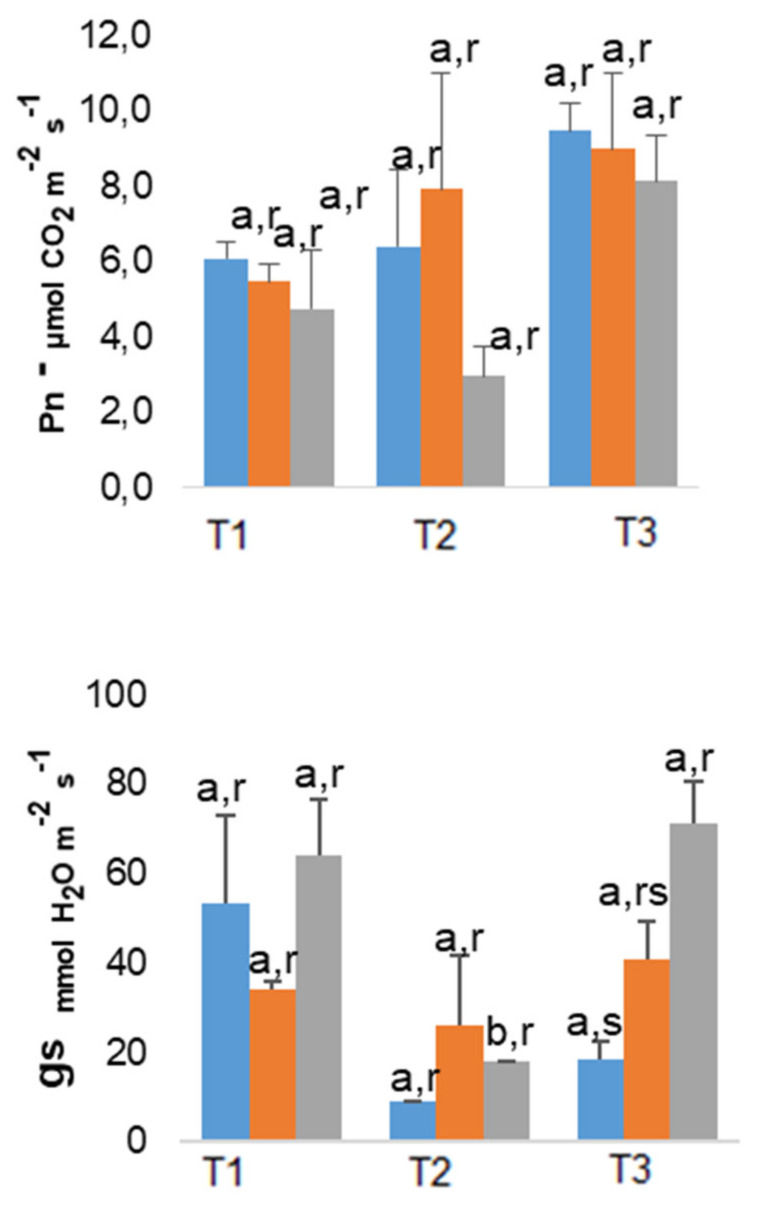
Variation in net photosynthesis rate (P_n_), stomatal conductance to water vapor (g_s_), and transpiration rate (Tr) in *E. globulus* plants, throughout the experimental period (T1—March; T2—May; T3—July). The mean values ± standard error (n = 5) followed by different letters express significant differences over time (a, b) or between As treatments (r, s); control (**blue**), 100 As (**orange**), and 200 As (**grey**).

**Figure 3 plants-10-00627-f003:**
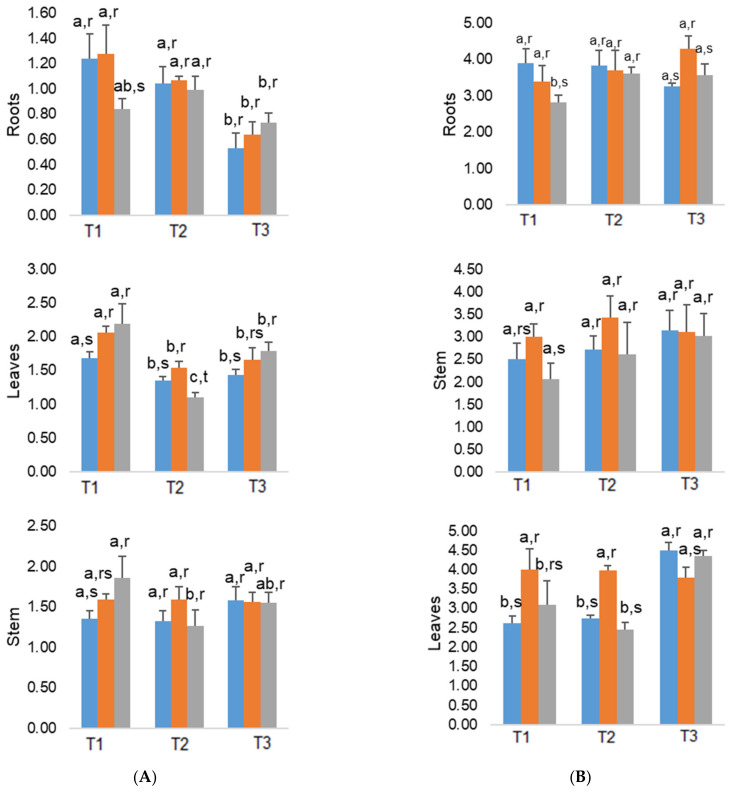
Average values of K (**A**) and Ca (**B**) in the roots, stems, and leaves of *E. globulus* As-treated plants, throughout the experimental assay. Different letters indicate significant differences at the 0.05 significance level. T1 = March; T2 = May; T3 = July; control (blue), 100 µg As mL^−1^ (orange), and 200 µg As mL^−1^ (grey).

**Figure 4 plants-10-00627-f004:**
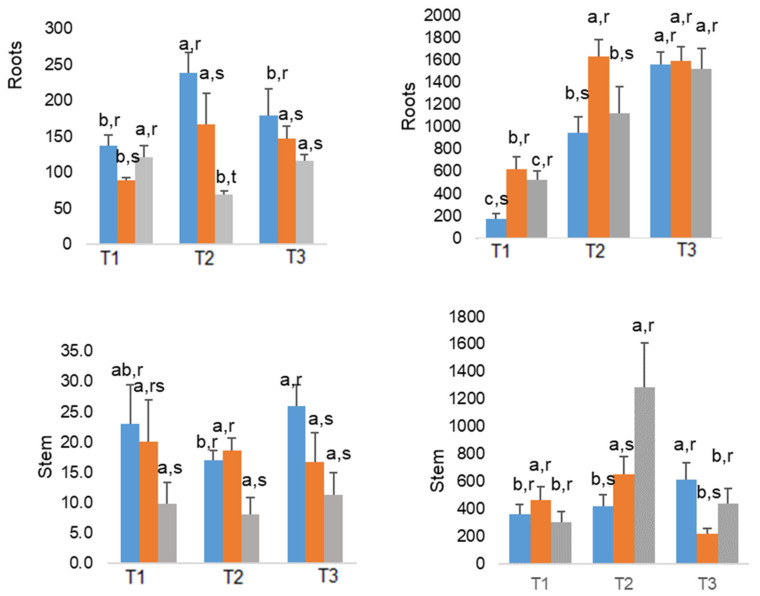
Average values of Zn (**A**) and Fe (**B**) in the roots, stems and leaves of *E. globulus* As-treated plants, throughout the experimental assay. Mean values are expressed as μg g^−1^ ± standard deviation (n = 3). Different letters indicate significant differences at the 0.05 significance level. T1 = March; T2 = May; T3 = July; control (**blue**), 100 As (**orange**), and 200 As (**grey**).

**Table 1 plants-10-00627-t001:** Variation of the content of As in different organs (μg g^−1^), total As (mg, obtained based on the As concentrations from the biomass), biomass (g dry weight), and plant height (cm) in *E. globulus* plants submitted to 100 or 200 µg As mL^−1^ treatments.

**March- T1**		Control	100 μg As mL^−1^	200 μg As mL^−1^
	As concentration Leaf	*BDL*	3.8 ± 0.3 ^a^	3.6 ± 0.6 ^a^
	Stem	*BDL*	*BDL*	*BDL*
	Root	*BDL*	49.2 ± 5.9 ^a,s^	74.3 ± 16.9 ^b,r^
	Total As (mg plant^−1^)	–	2.38	2.52
	Total biomass	92.7	148	86.2
	Leaf	37.8 ± 17.3 ^ab,r^	50.7 ± 19.0 ^a,r^	23.0 ± 6.1 ^b,s^
	Stem	39.5 ± 15.6 ^a,r^	50.3 ± 17.0 ^a,r^	33.3 ± 12.7 ^a,r^
	Root	15.4 ± 1.9 ^a,t^	47.4 ± 7.8 ^a,s^	30.9 ± 16.3 ^ab,s^
	Plant height	113 ± 4 ^a,r^	121 ± 3 ^a,r^	114 ± 10 ^a,r^
**May—T2**	As concentration Leaf	*BDL*	5.7 ± 0.5 ^b^	10.8 ± 0.4 ^a^
	Stem	*BDL*	*BDL*	*BDL*
	Root	*BDL*	52.9 ± 11.9 ^a,s^	102 ± 27.1 ^a,r^
	Total As (mg plant^−1^)	–	6.48	9.84
	Total biomass	167	202	194
	Leaf	32.0 ± 10.0 ^b,r^	42.8 ± 21.3 ^ab,r^	52.4± 6.0 ^a,r^
	Stem	41.4 ± 18.2 ^a,r^	41.1 ± 22.0 ^a,r^	51.0 ± 19.7 ^a,r^
	Root	93.6 ± 23.8 ^a,s^	118 ± 76.2 ^a,r^	91.0 ± 31.0 ^a,r^
	Plant height	133 ± 4 ^a,r^	120 ± 8 ^a,r^	134 ± 3 ^a,r^
**July—T3**	As concentration Leaf	*BDL*	6.7 ± 1.6 ^a^	4.4 ± 0.5 ^a^
	Stem	*BDL*	*BDL*	*BDL*
	Root	*BDL*	54.3 ± 28.7 ^a,s^	82.3 ± 22.1 ^ab,r^
	Total As (mg plant^−1^)	–	8.19	8.91
	Total biomass	317	235	196
	Leaf	40.2 ± 8.4 ^a,r^	39.6 ± 15.0 ^a,r^	42.9 ± 18.8 ^a,rs^
	Stem	66.1 ± 13.8 ^a,r^	49.8 ± 23.6 ^a,r^	46.8 ± 24.0 ^a,r^
	Root	211 ± 36 ^a,r^	146 ± 59 ^ab,r^	106 ± 47.6 ^b,r^
	Plant height	136 ± 8 ^a,r^	133 ± 7 ^a,r^	138 ± 5 ^a,r^

For each parameter, different letters after the mean values ± standard deviation (n = 3) express significant differences over time (^a,b^) or between As treatments in each date (^r,s^); *BDL* = below the detection limit.

**Table 2 plants-10-00627-t002:** Variation of leaf chlorophyll *a* fluorescence parameters in *E. globulus* plants submitted to 100 or 200 µg As mL^−1^ treatments, along the experiment.

Treatment	March (T1)			May (T2)			July (T3)		
				**F_o_**					
**Control**	0.13	±	0.01 ^b,r^	0.21	±	0.01 ^a,r^	0.23	±	0.03 ^a,r^
**100 As**	0.13	±	0.01 ^b,r^	0.19	±	0.03 ^a,r^	0.23	±	0.02 ^a,r^
**200 As**	0.14	±	0.00 ^b,r^	0.12	±	0.01 ^b,s^	0.23	±	0.04 ^a,r^
				**F_v_/F_m_**			**cv**		
**Control**	0.77	±	0.01 ^a,r^	0.77	±	0.04 ^a,r^	0.76	±	0.03 ^a,r^
**100 As**	0.78	±	0.03 ^a,r^	0.77	±	0.02 ^a,r^	0.76	±	0.02 ^a,r^
**200 As**	0.74	±	0.04 ^a,r^	0.77	±	0.02 ^a,r^	0.76	±	0.04 ^a,r^
				**Y_(II)_**					
**Control**	0.13	±	0.04 ^a,r^	0.17	±	0.02 ^a,rs^	0.17	±	0.01 ^a,r^
**100 As**	0.10	±	0.04 ^b,r^	0.24	±	0.03 ^a,r^	0.14	±	0.03 ^ab,r^
**200 As**	0.17	±	0.06 ^a,r^	0.15	±	0.02 ^a,s^	0.12	±	0.01 ^a,s^
				**Y_(NPQ)_**					
**Control**	0.55	±	0.01 ^a,r^	0.71	±	0.02 ^a,r^	0.66	±	0.01 ^a,rs^
**100 As**	0.52	±	0.12 ^a,r^	0.59	±	0.05 ^a,s^	0.65	±	0.02 ^a,s^
**200 As**	0.50	±	0.02 ^b,r^	0.64	±	0.04 ^b,s^	0.71	±	0.03 ^a,r^
				**Y_(NO)_**					
**Control**	0.32	±	0.03 ^a,r^	0.12	±	0.01 ^b,s^	0.17	±	0.02 ^b,r^
**100 As**	0.38	±	0.11 ^a,r^	0.17	±	0.03 ^b,rs^	0.21	±	0.01 ^b,r^
**200 As**	0.34	±	0.08 ^a,r^	0.22	±	0.03 ^b,r^	0.18	±	0.03 ^b,r^
				**q_N_**					
**Control**	0.85	±	0.02 ^b,r^	0.94	±	0.01 ^a,r^	0.92	±	0.01 ^a,r^
**100 As**	0.82	±	0.09 ^a,r^	0.88	±	0.04 ^a,r^	0.91	±	0.02 ^a,r^
**200 As**	0.83	±	0.03 ^b,r^	0.89	±	0.03 ^ab,r^	0.92	±	0.01 ^a,r^
				**q_L_**					
**Control**	0.33	±	0.07 ^a,r^	0.44	±	0.06 ^a,r^	0.51	±	0.05 ^a,r^
**100 As**	0.23	±	0.04 ^a,r^	0.56	±	0.18 ^a,r^	0.44	±	0.07 ^a,r^
**200 As**	0.44	±	0.07 ^a,r^	0.36	±	0.03 ^a,r^	0.33	±	0.01 ^a,r^
				**F_v_’/F_m_’**					
**Control**	0.32	±	0.03 ^a,r^	0.33	±	0.09 ^a,r^	0.28	±	0.04 ^a,r^
**100 As**	0.32	±	0.07 ^a,r^	0.39	±	0.04 ^a,r^	0.28	±	0.06 ^a,r^
**200 As**	0.32	±	0.33 ^a,r^	0.33	±	0.05 ^a,r^	0.30	±	0.02 ^a,r^
				**F_s_/F_m_’**					
**Control**	0.87	±	0.04 ^a,r^	0.83	±	0.02 ^a,rs^	0.84	±	0.01 ^a,r^
**100 As**	0.90	±	0.04 ^ab,r^	0.76	±	0.03 ^b,s^	0.86	±	0.03 ^a,r^
**200 As**	0.83	±	0.06 ^a,r^	0.85	±	0.02 ^a,r^	0.88	±	0.01 ^a,r^
				**Pl_Chr_**					
**Control**	7.12	±	3.29 ^a,r^	5.54	±	5.50 ^a,r^	7.28	±	3.09 ^a,r^
**100 As**	5.16	±	3.44 ^a,r^	6.69	±	2.47 ^a,r^	7.33	±	2.54 ^a,r^
**200 As**	10.3	±	5.9 ^a,r^	7.18	±	2.30 ^a,r^	7.12	±	5.40 ^a,r^

For each parameter, different letters after the mean values ± standard error (n = 8) express significant differences over time (^a,b^) or between As treatments in each date (^r,s^).

## Data Availability

Not applicable.

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
