# Peer review of "The Tolerance of Eucalyptus globulus to Soil Contamination with Arsenic"

_plants, 2021, doi:10.3390/plants10040627_

Round 1

Reviewer 1 Report

The article entitled “THE TOLERANCE OF Eucalyptus globulus TO SOIL CONTAMINATION WITH ARSENIC" presents the results of a study which aimed to determine the phytoremediation capability of Eucalyptus globulus Labilli, grown in arsenic contaminated soils during 6 months, in order to evaluate As uptake, accumulation and translocation efficiencies, and the effects on photosynthetic performance, as well as the possible effects on the uptake of other important micro and macronutrients to the plant.

The research carried out is very interesting and in line with the current trends. Recently, research on arsenic, its sources and negative effects on plants has become very popular. The data presented are of more than national interest.

However, there is a need to improve and/or clarify some aspects.

General note. The results are properly described, the discussion is interesting. In my opinion, the way of presenting the results in tables, and especially in figures, requires improvement.

Introduction

Phosphorus fertilizers are used in many countries, not only in the Sri Lanka, and it seems that the greater problem in the use of phosphorus fertilizers is the incorporation of cadmium into the soil, not arsenic.

Results

Arsenic accumulation

Table 1. for some values, no superscripts.

The layout of the table is a bit confusing. Results for Leaf, Stem, Root are given twice for each test term (?)

Plant growth and foliar traits

Wrong font format (italics).

Photosynthetic related parameters

Figure 1. Wrong font format (italics).

It would be better to put a legend next to the figures than to write in the descriptions under the figures which means the colors on the figures. I propose to change this on all figures

Presented in the figure The significance of the differences between As treatment is strange. For example, the SLA parameter in T1 does not differ between control treatment and As 100 and 200. Meanwhile, for As 100 the SLA result is higher.

Figure 2. change the figure format to the same as the other figures

Figure 3. Some figures were moved while formatting the text. Check it and correct it. Please add variants “a” and “b” above the graphs with the description in the legend that “a” is for potassium. The applied right / left method is not very popular in scientific articles

Figure 4. note as for figure 3

Discussion

Accumulation of macro (Ca and K) and micronutrients (Fe and Zn) in As treated plants

The authors state "The four studied elements are essential to plant nutrition". This is not true, especially in relation to the tested micronutrients. Besides, nitrogen is the basic and most important nutrient for plants.

no superscript at Ca+2

Materials and Methods

Plant material and experimental design

P2O5: 80- 150 mg L-1 ; K2O: 300-500 mg L-1 ; these data should be converted into P and K, since the scientific articles report the content of phosphorus and potassium in the elemental form rather than in the form of oxides.

Conclusions

Why is one paragraph of conclusions in italics?

References

In the article, the authors used a large amount of literature (84 items). However, attention is drawn to the fact that a large part of the cited literature is older than 10 years.

Author Response

 A point-by-point response to the reviewer’s comments is in attach

Reviewer 2 Report

The tolerance of Eucalyptus globulus to soil contamination with Arsenic. Cite several old references and has a poor sentence structure, especially in the discussion. Besides that, the work is quite useful and suitable for publication.

Author Response

 a point-by-point response to the reviewer’s comments is in attach

Reviewer 3 Report

Dear Authors,

This manuscript can be considered for possible publication if the authors can address the below-mentioned comments for further improvement.

  • Please start the abstract with the introductory lines, problem, objectives, followed by the methodology, etc.
  • There are several undefined abbreviations within the abstract. Please define them on the first appearance. Also, check the entire text for undefined abbreviations.
  • Please be consistent with arsenic or As. The authors have used both terms. I suggest defining it on the first appearance in the introduction and use the abbreviated form throughout the text. Further, please avoid starting a sentence with an abbreviation.
  • The second line of the introduction, “in our country, " better mentions the country name instead of “our”.
  • Plants 5 months old of Eucalyptus globulus Please change it to Five months old Eucalyptus globulus plants.
  • Please do not add all the formulas within the text. Present them on the next line in terms of formula/equation and define them. For example, sections 4.2, 4.4.
  • Section 2.2. and conclusion, please check if the text should be italic or not.
  • For fig 3, please arrange all the graphs in one image. Avoid extra spaces in fig 3 and 4.
  • Section 4.7, how about replications?
  • There are some language errors throughout the text.

Author Response

the response to the reviewer is in attach

Round 2

Reviewer 1 Report

I appreciate the  authors' efforts on this manuscript, which indeed improve the quality of this manuscript. Particularly,  the authors added missing information, updated data, and improved Figures. Thus, I satisfy the authors' respondence and the revision.

Author Response

Thank you again for your precious help

Best regards

FR

Reviewer 2 Report

The authors have improved the manuscript, and therefore, it can be accepted for publication.

Author Response

(The authors gave the same response as above.)

Reviewer 3 Report

Dear Authors,

Thank you for revising the MS according to the proposed comments and suggestions. Notably, the MS has been improved and can be accepted after minor revision.

  • The second paragraph of the conclusion is still italic. Please make sure it should not be in italic in the final version before publication.
  • Section 4.7, though there is a sign of 3n replications in figures and table. This fact mainly belongs to the methodology. So, I strongly advised writing a line and mention that three biological/technical replication was used. Of course, this is just for your consideration.
  • Figures-no did not mean to separate the figures. There are many extra spaces between each graph and half on the next page and half on the previous page. Thus, I suggest arranging them on one page or one PNG file (individual figure). If no, please pay attention to the proof and work closely with the production team to avoid any graph position mistakes. Thanks.

Author Response

Dear Reviewer

Regarding the 2nd paragraph of the conclusions, originally it was not in italic. I corrected the style

As requested in the item 4.7 it was added the number of replications in each case.

I agree with you that we must pay attention to the proofs and work closely with the production team to avoid any graph position mistakes. Working in the current format I detect that in Figure 2 the gs data was missing.

Thank you again for your precious help

Best regards

FR